# Sulfite preservatives effects on the mouth microbiome: Changes in viability, diversity and composition of microbiota

**Sally V. Irwin**[1][⊕]*, **Luz Maria Deardorff**[2‡], **Youping Deng**[3‡], **Peter Fisher**[1⊕],
**Michelle Gould**[1⊕], **Junnie June**[1⊕], **Rachael S. Kent**[1⊕], **Yujia Qin**[3‡], **Fracesca Yadao**[1‡]

**1** Department of Science, Technology, Engineering and Mathematics, University of Hawai'i Maui College, Kahului, Hawai'i, United States of America, **2** Department of Natural Sciences, University of Hawai'i at Manoa, Honolulu, Hawai'i, United States of America, **3** Department of Quantitative Health Sciences, John A. Burns School of Medicine, University of Hawaii at Manoa, Honolulu, Hawai'I, United States of America

⊕ These authors contributed equally to this work.
‡ These authors also contributed equally to this work.
* sirwin@hawaii.edu

**Data Availability Statement:** Data is available in manuscript, supplemental information and in the NCBI BioProjectdatabase https://www.ncbi.nlm.nih.gov/bioproject/PRJNA766452.

## Abstract

### Overview

Processed foods make up about 70 percent of the North American diet. Sulfites and other food preservatives are added to these foods largely to limit bacterial contamination. The mouth microbiota and its associated enzymes are the first to encounter food and therefore likely to be the most affected.

### Methods

Eight saliva samples from ten individuals were exposed to two sulfite preservatives, sodium sulfite and sodium bisulfite. One sample set was evaluated for bacteria composition utilizing 16s rRNA sequencing, and the number of viable cells in all sample sets was determined utilizing ATP assays at 10 and 40-minute exposure times. All untreated samples were analyzed for baseline lysozyme activity, and possible correlations between the number of viable cells and lysozyme activity.

### Results

Sequencing indicated significant increases in alpha diversity with sodium bisulfite exposure and changes in relative abundance of 3 amplicon sequence variants (ASV). Sodium sulfite treated samples showed a significant decrease in the Firmicutes/Bacteroidetes ratio, a marginally significant change in alpha diversity, and a significant change in the relative abundance for Proteobacteria, Firmicutes, Bacteroidetes, and for 6 ASVs. Beta diversity didn't show separation between groups, however, all but one sample set was observed to be moving in the same direction under sodium sulfite treatment. ATP assays indicated a significant and consistent average decrease in activity ranging from 24–46% at both exposure times with both sulfites.

**Funding:** SVI. PF. MG,JJ, RK, LD, FY This project was supported by grants from the National Institutes of Health (NIH), National Institute of General Medical Sciences (NIGMS), IDeA Networks of Biomedical Research Excellence (INBRE), Award number: P20GM103466. The content is solely the responsibility of the authors and do not necessarily represent the official views of the National Institutes of Health. https://www.nigms.nih.gov/research/drcb/IDeA/Pages/INBRE.aspxThe funders had no role in study design, data collection and analysis, decision to publish, or preparation of the manuscript. YD, YQ This work is partially supported by the NIH grants 5P30GM114737, 5P20GM103466, 5U54MD007601 and 5P30CA071789 https://grants.nih.gov/funding/index.htm The funders had no role in study design, data collection and analysis, decision to publish, or preparation of the manuscript.

**Competing interests:** The authors have declared that no competing interests exist.

Average initial rates of lysozyme activity between all individuals ranged from +/- 76% compared to individual variations of +/- 10–34%. No consistent, significant correlation was found between ATP and lysozyme activity in any sample sets.

## Conclusions

Sulfite preservatives, at concentrations regarded as safe by the FDA, alter the relative abundance and richness of the microbiota found in saliva, and decrease the number of viable cells, within 10 minutes of exposure.

## Introduction

The human oral cavity is a complex environment hosting up to 700 different species of bacteria, found primarily in 5 phyla, residing in saliva, teeth surfaces and on the apical mucosa of the tongue and cheeks [1–3]. At birth, colonization of the mouth begins. *Streptococcus salivarius*, a facultative anaerobe, is one of the initial colonizers along with several aerobes within the 1st year, including *Lactobacillus*, *Actinomyces*, *Neisseria* and *Veillonella*, an anaerobe. [4]. Diversity and the number of bacteria present in different individuals' mouths may vary as a result of environmental and genetic factors including disease and diet. However, recent studies have found most healthy individuals maintain a fairly consistent population over weeks to months of time [2,5].

Human saliva contains several enzymes and buffers that contribute to the first steps in digestion and serve as a first line of defense in immunological responses [6,7]. Lysozyme is found in saliva, as well as tears, blood serum, perspiration, and other bodily fluids. It is an antimicrobial enzyme that catalyzes cleaving of ß(1,4)-glycosidic bonds between residues of N-acetylmuramic acid (NAM) and N-acetylglucosamine (NAG) in peptidoglycan of primarily gram positive bacterial cell walls. Research has shown that lysozyme found in the mouth can lower the adherence of bacteria to surfaces and limit the number of microbes [8,9].

An imbalance in the mouth microbiome can lead to oral diseases such as dental caries, periodontitis, oral mucosal disease, and systemic diseases of the gastrointestinal, cardiovascular, and nervous systems [1,10]. A few recent studies have detected significant changes in the mouth microbiota in response to short and long-term environmental alterations [11–14]. Bescos *et.al.* observed the effects of chlorhexidine mouthwash resulting in a significant increase in the percent abundance of *Firmicutes* and *Proteobacteria* and a decrease in *Bacteroidetes*, *Fusobacteria*, TM7, and SR1 [11]. The effects of inorganic nitrates (found in vegetables) on the mouth microbiome of 19 omnivores and 22 vegetarians, displayed an increase in the percent abundance of *Proteobacteria* and a decrease in *Bacteroidetes* consistent with the chlorhexidine mouthwash study [12]. Another study, examining the effects of green tea on both the gut and mouth microbes showed "an irreversible increase of *Firmicutes* to *Bacteroidetes* ratio, elevated SCFA producing genera, and reduction of bacterial LPS synthesis in feces" [13]. A study examining betel nut chewing in 122 individuals, found reduced bacterial diversity, elevated levels of *Streptococcus infantis*, and changes in distinct taxa of *Actinomyces* and *Streptococcus* genera in current users and reduced levels of *Parascardovia* and *Streptococcus* in long-term chewers [14]. At the phyla level in both the mouth and gut, an increase in Firmicutes is often observed with obesity, and an increase in Bacteroidetes is observed with inflammatory bowel disease [10,15,16]. Proteobacteria, another key player in the oral microbiome, has been found to be

the most variable phylum in dysbiosis and its increase is generally related to a decrease in Firmicutes [10,16].

Antibiotics, diet, and food additives including artificial sweeteners, emulsifiers, preservatives, colorants, and acidity regulators can result in dysbiosis in the gut leading to a variety of human health issues [17–20]. An unhealthy gut microbiome influences signaling in the gut-brain axis; a bidirectional signaling system responsible for homeostasis, function, and overall health [21]. A recent study in germ-free humanized mice found that a mixture of common antimicrobial food additives including sodium benzoate, sodium nitrite, and potassium sorbate induced a significant dysbiosis in the gut [22]. Sodium sulfite and sodium bisulfite have long been used as preservatives to prevent food spoilage and browning. They can be found in a variety of foods such as dried fruit, processed meat, beer, wine, and canned goods [23]. Many foods contain such high levels of sulfites that exposure above levels generally regarded as safe (GRAS) by the US Food and Drug Administration (FDA) is common [23]. Our previous study demonstrated that under ideal growing conditions, sodium sulfite at 3780ppm and sodium bisulfite at 1800 ppm (both lower than the 5000ppm allowable for GRAS), were bactericidal to four probiotic species in 2–6 hours of exposure [24]. Few studies have been conducted to determine the effect of preservatives on the gut microbiota and none are known to us regarding their impact on the mouth microbiome.

Most recent studies on the human mouth and gut microbiota have utilized 16s rRNA sequencing, and in some cases flow cytometry or CFUs along with OD600 spectrophotometry to quantify cell numbers [2,5]. Twenty to sixty percent of the bacterial cells in the mouth microbiome are estimated to be unculturable, making plate counts and CFUs of limited use [2]. This has made 16S rRNA sequencing a powerful tool to profile mixed communities of bacteria. However immediate or short-term changes in the microbiome cannot reliably be detected due to the complex nature of the communities and the detection of DNA sequences from both living and recently lysed cells [2]. Flow cytometry is thought to be the gold standard for cell counting but also has limitations due to the potential presence of bacterial aggregates which may artificially deflate the cell count [2].

The concentration of adenosine triphosphate (ATP) present in living cells can be quantified using a proportional luminescent signal to observe immediate changes in the numbers of viable bacterial cells in a sample. Sensitivity as low as 0.0001nM ATP, reflecting the test population, is high due to the rapid loss of ATP in non-living cells [25]. This method implements a recombinant luciferase enzyme to oxidize luciferin in the presence of ATP and oxygen to produce oxyluciferin and light. It has been used to accurately quantify changes in the number of viable cells in various antibiotic susceptibility tests ranging from in-vitro biofilm formations by *Pseudomonas aeruginosa* [26], multiple gram-negative bacterial species within 24 hours of treatment with antibiotics [27], and slow growing species such as *Borrelia burgdorferi* [28]. Limitations to this method include the detection of eukaryotic cell's ATP present in some mixed populations, differences in amounts of ATP produced by varying cell types and inhibition of the luminescence signal due to the media used in assays. All these factors were considered in the experiments described here.

This study examines the effects of two types of sulfite preservatives on the human mouth microbiome by corresponding changes in ATP activity of saliva samples collected over a 4–5-week period from 10 individuals. Bacteria in saliva samples from one individual were examined for responses to the sulfite treatments indicated by changes in the percent abundance and/or diversity with 16s rRNA sequencing. Additionally, baseline lysozyme activity in the individuals being studied was determined and compared to the number of viable cells found in the samples.

## Methods and materials

### Saliva collection

Unstimulated saliva was collected from 10 individuals between 2–10 hours after consuming food or drink, other than water. Prior to donation, each individual rinsed mouth 3 times with water followed by a twenty-minute waiting period before delivering saliva via spitting into a sterile tube. Samples were placed on ice for the duration of collection and aliquoting. Time allowed for saliva collection was kept under 30 minutes. Collections took place twice a week with at least 2 days in between, over 4–5 weeks. Each participant supplied 8 samples total. Collected saliva was diluted 1:10 with sterile DI water, aliquoted and stored at -80°C. Participants contributing to saliva collections included ten individuals, 2 males (M1 and M2) and 8 females (F1-F8) from 18–60 years of age, all in overall good health. Our protocol for utilizing human saliva samples was reviewed and permitted by the Institutional Biosafety Committee (IBC) at the University of Hawaii. The Internal Review Board was also consulted and it was determined that no further review was necessary because the samples were made anonymous immediately after collecting, and they were not considered human samples due to the analysis performed being directed only on the bacterial cells. Aliquots of the same samples were used in all experiments.

### Sulfite exposure to saliva samples

Saliva aliquots were exposed to freshly made, 1800 ppm (17.3 mM) sodium bisulfite (Fisher Scientific) or 3780 ppm (30 mM) anhydrous sodium sulfite (Fisher Scientific) both prepared in sterile water in independent experiments. Immediately following exposure to sulfites, aliquots of each control and matching exposed sample were centrifuged at 4,600 RCFs (7000 RPMs) for 10 minutes to pellet cells. The supernatant was discarded and the pellet brought up in an equal volume of 1X phosphate buffered saline (PBS). These samples are referred to as "time 0" (T0). The other half of each aliquot was placed in an incubator set at 36°C for 30 minutes (T30), then pelleted using the same procedure described for time zero samples, brought up in 1X PBS and stored. All samples were stored at -20°C for short term storage of 6 weeks or less, or at -80°C for longer storage. See supplemental materials for additional notes on procedure.

### ATP activity in saliva samples

BacTiter-Glo™ Microbial Cell Viability Assay from Promega was used to quantify relative numbers of viable bacterial cells found in saliva samples. The Perkin Elmer Victor X3 multi-mode plate reader was used to record luminescence of all samples. Samples were assayed in triplicate using a white plate with clear bottom (View-Plate 96 TC from PerkinElmer). Lids were coated with Triton X100 as described by Brewster [29] to prevent condensation.

Blank wells were filled with sterile water to cut down on cross talk, negative controls contained 100ul of 1X PBS + 100ul BacTiter-GLO™ reagent. One hundred microliters of each treated and control (untreated) sample (3 replicates each) were tested. All samples from each individual were ATP assayed on the same day using the same Bac-titer glo reagent to limit variability between luminescence readings for more accurate comparisons of results. Samples and BacTiter-Glo™ reagent were brought to room temperature before use. Assay consisted of a 5 second orbital (0.10mm) shake, a 5-minute incubation at 25°C, followed by a 1 second read of luminescence (RLU).

### Statistical analysis of ATP studies

For all statistical tests, p values < 0.05 were considered significant. Statistical errors were calculated as the standard error (SE) or standard deviation (SD). Average total RLU (relative light

units) for each individual's eight saliva samples were assessed using raw RLU data by a two-way ANOVA and paired T-Tests.

## Sample preparation for DNA extraction

A total of 32 saliva samples from individual F2 was sent to Zymo Research, Irvine, CA for 16s rRNA sequencing. Sequenced samples included 8 sodium sulfite treated saliva samples at T0, and 8 untreated saliva samples serving as controls at T0. Eight sodium bisulfite treated saliva samples at T0 and the respective control samples. A 10:1 RNA/DNA shield from Zymo was added to each saliva sample to maintain samples during shipping.

## DNA extraction

DNA was extracted from saliva samples using ZymoBIOMICS®-96 MagBead DNA Kit (Zymo Research, Irvine, CA). The ZymoBIOMICS® Microbial Community Standard was used as a positive control. A blank extraction control was used as the negative control.

## Targeted library preparation

Bacterial 16S ribosomal RNA gene targeted sequencing was performed using the *Quick*-16S™ NGS Library Prep Kit (Zymo Research, Irvine, CA). Bacterial 16S primers amplify the V3-V4 region of the 16S rRNA gene. The ZymoBIOMICS® Microbial Community DNA Standard was used as a positive control. A blank library preparation was used as the negative control.

## Absolute abundance quantification

Zymo Research indicates the following methods were used for absolute abundance quantification: A quantitative real-time PCR was set up with a standard curve. The standard curve was made with plasmid DNA containing one copy of the 16S gene and one copy of the fungal ITS2 region prepared in 10-fold serial dilutions. The primers used were the same as those used in Targeted Library Preparation. The equation generated by the plasmid DNA standard curve was used to calculate the number of gene copies in the reaction for each sample. The PCR input volume was used to calculate the number of gene copies per microliter in each DNA sample.

The number of genome copies per microliter DNA sample was calculated by dividing the gene copy number by an assumed number of gene copies per genome. The value used for 16S copies per genome is 4. The value used for ITS copies per genome is 200. The amount of DNA per microliter DNA sample (DNA_ng) was calculated using an assumed genome size of $4.64 \times 10^6$ bp, the genome size of *Escherichia coli*, for 16S samples. This calculation is shown below:

Calculated Total DNA = Calculated Total Genome Copies × Assumed Genome Size $(4.64 \times 10^6$ bp$)$ × Average Molecular Weight of a DNA bp (660 g/mole/bp) ÷ Avogadros Number $(6.022 \times 10^{23}$/mole$)$. (Zymo Research, Irvine, CA).

## Sequencing

The final library was sequenced on Illumina® MiSeq™ with a v3 reagent kit (600 cycles). The sequencing was performed with 10% PhiX spike-in.

## Bioinformatics analysis

Unique amplicon sequence variants (ASVs) were inferred from raw reads using the DADA2 pipeline as described by Callahan et al. [30]. Chimeric sequences were also removed with the

DADA2 pipeline. Taxonomy assignment was performed using Uclust from Qiime v.1.9.1 with the Zymo Research Database.

Firmicutes/Bacteroidetes ratio was calculated using the number of sequences belonging to phylum Firmicutes divided by the number of sequences belonging to Bacteroidetes. To evaluate the significance of the differences between the control and treatment groups, paired t-tests were used for statistical analysis, and the results (with p-value < 0.05) shown in the figures. Alpha diversities were estimated using four indexes using R package "Vegan" [31]: including species number (richness), Chao1, Shannon index, and Inverse Simpson index. The beta diversity can be explained by the PCA (principal component analysis) plots, which was also conducted by the R package "Vegan" using the Bray-Curtis distance matrix at the ASV level. The relative abundances of ASVs in different samples were transformed and the first two principal components were plotted to show the relationships between the groups. The percentage followed in the axis shows the portion of the total variances that can be explained by the first or the second principal component. The differentially abundant taxa (at different phylogenetic levels) were identified based on the paired t-test results.

### Lysozyme activity assays

Lysozyme activity initial rates were determined for all individuals untreated samples, using a Perkin Elmer Victor X3 multimode plate reader and an EnzChek™ Lysozyme Assay Kit (E-22013) from ThermoFisher Scientific at a temperature of 37˚C. Black 96 well TC ViewPlates (PerkinElmer) with lids coated with Triton X100 as described by Brewster [29] to prevent condensation. Fifty microliters of saliva samples, 100ul of ENZ Check buffer, and 50ul of substrate was added to each well in triplicate.

### Graphing

Results were analyzed and graphed using Originlab software.

## Results

### ATP assays

Two-way ANOVA tests with replication using raw (RLU) data indicate that there is no significant interaction between specific individuals and their reactions to sulfite treatments or between individual's reactions to sulfite treatment due to exposure time. However, there is a significant difference in ATP levels between individual's control and treated saliva samples, with a large variation in baseline numbers of bacteria present in each person's mouth and a significant difference in ATP levels between sulfite treated and control samples for all individuals (Table 1, Fig 1 and S2 Appendix).

**Baseline lysozyme and comparison to number of viable cells (ATP).** A baseline of lysozyme activity in the collected saliva samples was established over a five-week period. Activity was determined as a function of initial rates of ten individuals (Figs 2 and 3). A one-way ANOVA indicated an overall significant difference between individuals of p = 5.5 E-15. Tukey comparisons indicated in Fig 2 show each sample's range and similarity among samples. A large variation in the average initial rates over the five weeks of collection time between all individuals was observed, ranging from +/- 76% compared to individual initial rate variations of +/- 34%. No consistent correlation between viable cells (ATP) and Lysozyme activity was observed (Fig 3).

**Table 1. "p-value" results of two-way ANOVA tests.**

| | Time 0 | | | Time 30 | | |
|---|---|---|---|---|---|---|
| | Individuals | Treatment | Interaction | Individuals | Treatment | Interaction |
| SodiumSulfite | 6.38E-37*** | 0.009** | 0.500 | 2.37E-31*** | 0.035* | 0.997 |
| SodiumBisulfite | 1.06E-28*** | 0.002** | 0.121 | 2.41E-28*** | 1.35E-05*** | 0.065 |

Results show comparison of individual's reactions to sulfite treatments, sulfite treatments compared to exposure time, and treated vs control samples for all saliva samples.

(* $p < 0.05$

** $p < 0.01$

*** $p < 0.001$).

## Comparison of individual F2 ATP, lysozyme activity and 16s rRNA sequencing

A significant decrease in viable cells based on ATP activity (RLU) in F2's samples treated with $Na_2SO_3$ and $NaHSO_3$ was observed, with a 40% and 30% decrease (respectively) illustrated in Fig 4A and 4B.

## 16s rRNA sequencing results

Individual "F2's" samples indicated that Bacteroidetes, Proteobacteria and Firmicutes phyla significantly varied in relative abundance with $Na_2SO_3$ exposure (Figs 4A, 5A and 6A).

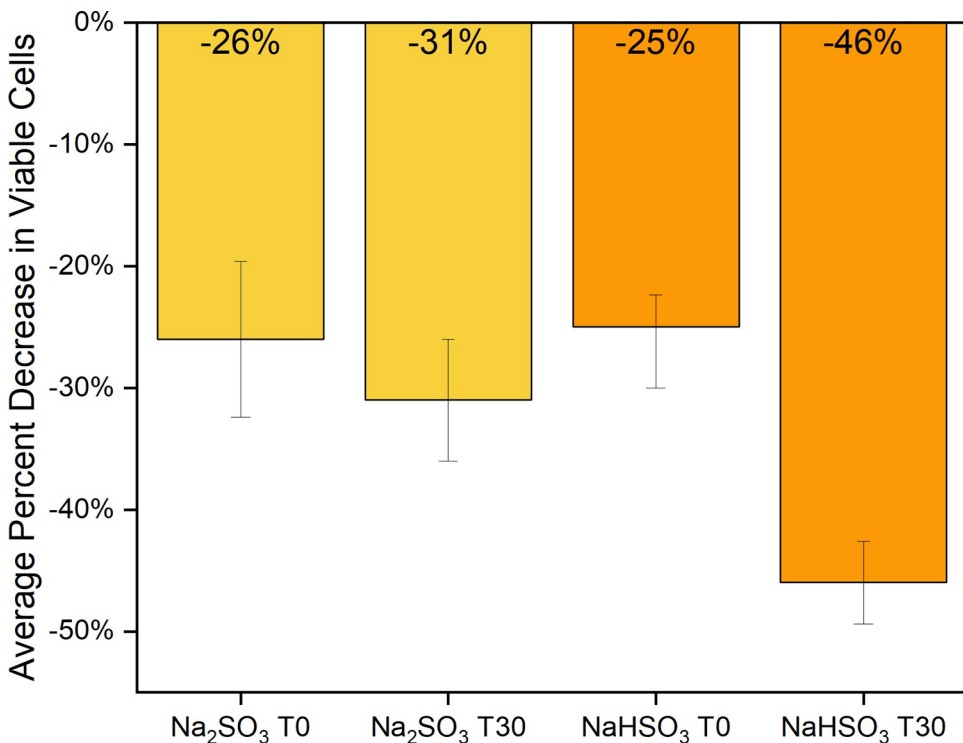

**Fig 1. Average change in ATP activity after sulfite treatment.** Raw data (RLU) of all saliva samples (F1-F8 and M1-M2) treated with $Na_2SO_3$ or $NaHSO_3$ compared to controls was used to determine the average % decrease of viable cells (based on ATP activity) in each at time 0 (T0) and time 30 (T30). The standard error of each test group was calculated, and evidence of observed difference was confirmed by both the paired t-test and the two-way ANOVA (Table 1).

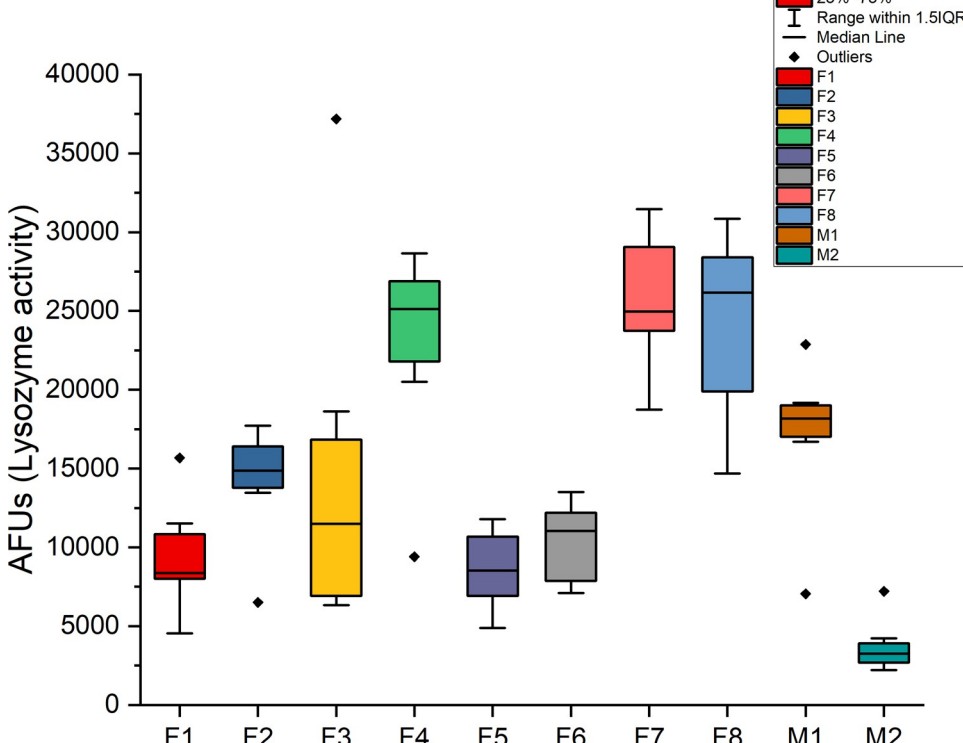

**Fig 2. Lysozyme activity in untreated saliva samples.** Untreated saliva samples from all individuals (n = 10) were assayed for initial rates of lysozyme activity. The average initial rate from replicates of 3 from each sample set/individual (n = 8) were used to determine the standard deviation and median scores.

However, no significant difference in the relative abundance of the top 5 phyla were observed with $NaHSO_3$ (Figs 4B and 5B).

At a more detailed level, significant changes to 3 genus and 6 ASV's (Fig 6A–6C) were also observed in the $Na_2SO_3$ treated samples and 3 ASV's in the $NaHSO_3$ assays also showed a significant change in relative abundance (Fig 7).

A summary of the bacterial types at the genus and ASV level is presented in Table 2. Nine different anaerobic ASV's exhibited significant changes in relative abundance with sulfite treatments. Four out of the five that showed an increase in abundance were Gram negative and three out of those five were sulfite reducing bacteria (SRB) or have the ability to avoid oxidative stress. Four ASV's decreased in relative abundance with sulfite treatments. Three out of the four were Gram positive and one out of the 4 was a SRB.

Alpha diversity measured by species richness and Chao 1 showed the same significant increase after exposure to $NaHSO_3$ and a marginally significant increase upon exposure to $Na_2SO_3$ (Fig 8). However, no significant differences were identified with the Shannon or Inverse Simpson index.

Baseline lysozyme activity for F2 samples compared to the respective number of viable cells (based on ATP activity) did not indicate a correlation of any type between these two parameters (S1 Fig).

## Beta diversity

The PCA plot shows consistent changes in the oral microbial communities when comparing the control samples to the $Na_2SO_3$ treated samples of F2. As shown in Fig 9, the dissimilarity

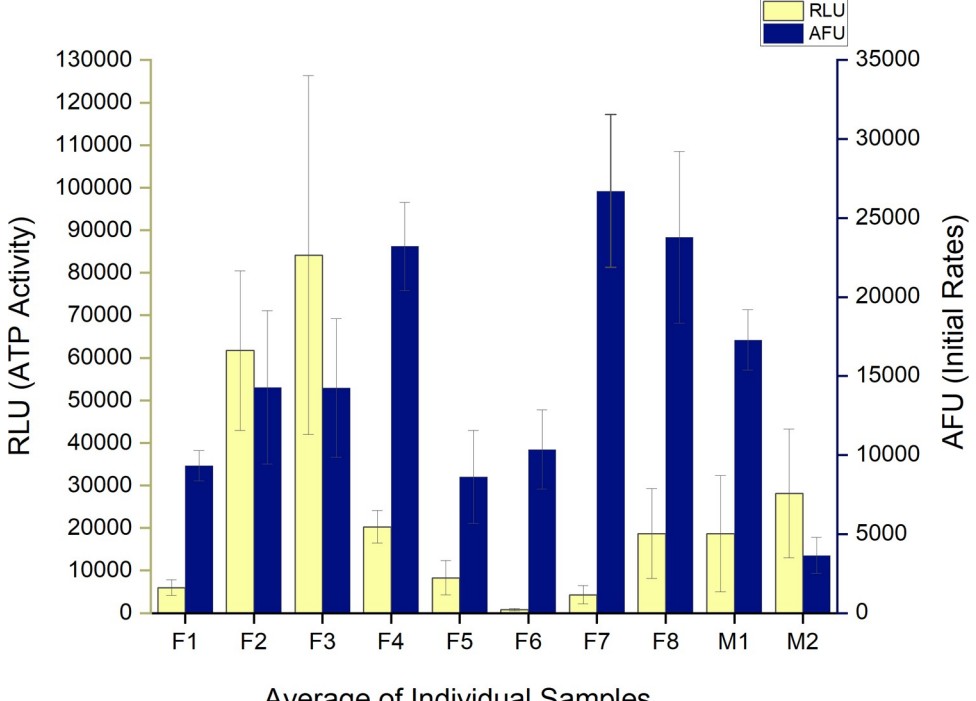

**Fig 3. Viable cells and lysozyme activity in control samples.** 'Comparison of averaged ATP activity and lysozyme initial rates in all 10 individual's control saliva samples. Standard deviation of averaged sample sets indicated for both RLU and AFU.

(beta-diversity) within the samples was large due to the possible micro-environment changes affected by many factors such as diet, hormonal status, etc. However, when treated with $Na_2SO_3$, the changes in the microbial community composition were towards almost the same direction, which indicates that the treatment of $Na_2SO_3$ caused consistent changes to the oral microbial community, even though their original structures were not similar.

## Discussion

Preservatives in food interact with the human microbiome first through the mouth and then the rest of the digestive tract. Sulfites are a common preservative added to a variety of foods and occur naturally in some fermented foods. Since being regulated as GRAS for use in food in 1958 by the US FDA, allowable amounts and applications have changed several times as it became clear that sulfites were causing moderate to severe (even fatal) health effects in some individuals [20,23,37,38] Mammals oxidize sulfite to sulfate with sulfite oxidase and some SRB use sulfate as an energy source, which may result in the production of $H_2S$ [39,40]. Both insufficient production of sulfite oxidase leading to an increase in several forms of sulfite reactive compounds, and oxidative stress from $H_2S$ on eukaryotic cells, may occur as the result of ingestion of sulfites. A recent study in rats showed that $Na_2SO_3$ ($\geqq$ 100ppm) causes death of gastric mucosal cells through oxidative stress [41]. This same study also looked at the effects of sodium sulfite (1mM) on lysozyme activity. They determined that $Na_2SO_3$ decreased the bacteriolytic activity of lysozyme in a time and dose dependent manner [41]. Other studies have shown a connection between certain types of SRB found in the colon and ulcerative colitis [42]. These studies provide evidence of mechanisms by which $Na_2SO_3$ may lead to both eukaryotic and prokaryotic cell death.

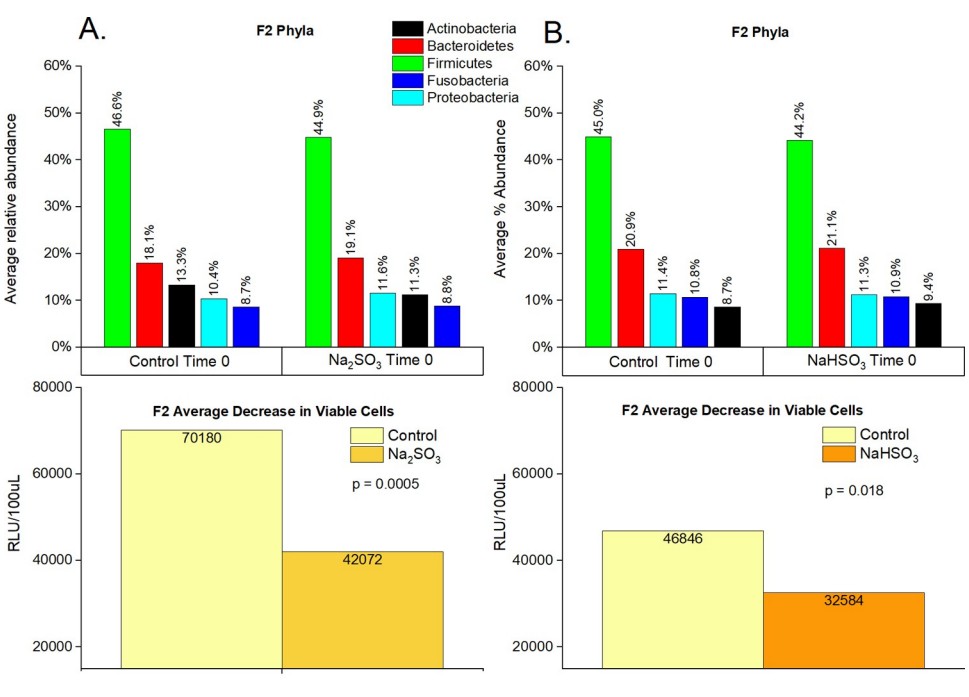

**Fig 4. Cell viability and relative abundance of phyla in F2 samples before and after sulfite treatment.** F2 saliva samples (n = 32) were sequenced (16S rRNA) and the average percent abundance of the most predominant 5 phyla is displayed for untreated controls and sodium sulfite (A) or sodium bisulfite (B) treated samples at time 0. "p-values" were calculated using paired t-tests.

Our previous work indicated that sodium sulfite was bactericidal to three probiotic species of *Lactobacillus* and bacteriostatic to *Streptococcus thermophilus*, within 2–6 hours of exposure [24]. Sodium bisulfite was found to be bactericidal to all 4 probiotic bacteria also within 2–6 hours of exposure [24]. These studies established both the potency of these preservatives and the varying susceptibility of different types of bacteria when exposed under ideal culture conditions. The current study was conducted to evaluate the effects of these same sulfites on the mouth microbiome found in saliva to better simulate the effects of ingesting foods containing these preservatives.

## ATP activity

Comparison of each individual's control samples indicated that the viable number of cells were relatively consistent over the 5-week sampling time, which supports findings from other recent studies [2,5,10]. The most striking observations were the differences found in ATP activity (indicating viable cell counts), between controls and sulfite treated samples at both exposure times. Based on these results we can conclude that both sodium sulfite and sodium bisulfite, consistently and immediately (regardless of preservative type or sample origin) significantly reduced the number of viable bacterial cells in saliva samples. We are assuming this is a drop in the number of viable cells, but it could be partially due to sulfites interfering with ATP production, or ATP being utilized by SRB to metabolize the sulfites.

These results were also supported by our preliminary experiments on the mouth microbiome which examined the effects of sulfites on 12 saliva samples from 5 individuals over a three-month period. Very similar results of reduced ATP activity as described here were

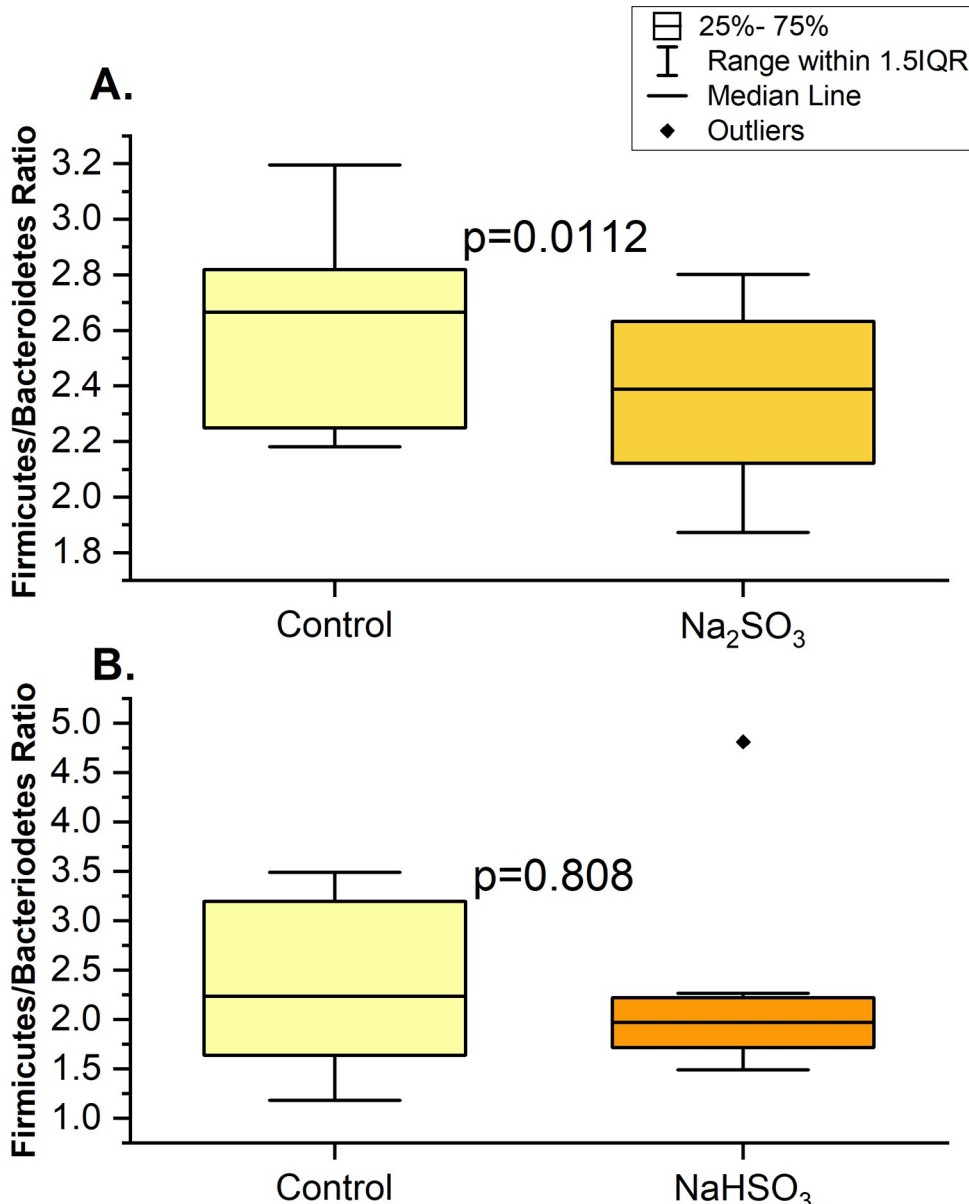

**Fig 5. Ratio of firmicutes/bacteroidetes changes in F2 samples with sulfite treatments at time 0.** 16S rRNA sequencing was used to determine the ratio (based on % abundance) for control and treated samples with $Na_2SO_3$ (A) or with $NaHSO_3$ (B) of individual F2.

observed, with an average decrease in viable cells of 27% and 28% at time 0 for $Na_2SO_3$ and $NaHSO_3$ respectively. Additional experiments observed changes in the microbial population in these saliva samples by recording OD600 readings over a 6.5-hour period. We observed an initial decrease in cell numbers in the treated which remained lower than controls throughout study, and an increase in cell numbers in the controls.

## Lysozyme

Lysozyme activity did not vary significantly over the 5 weeks of sample collection/individual and there did not appear to be any correlation between the number of viable cells and

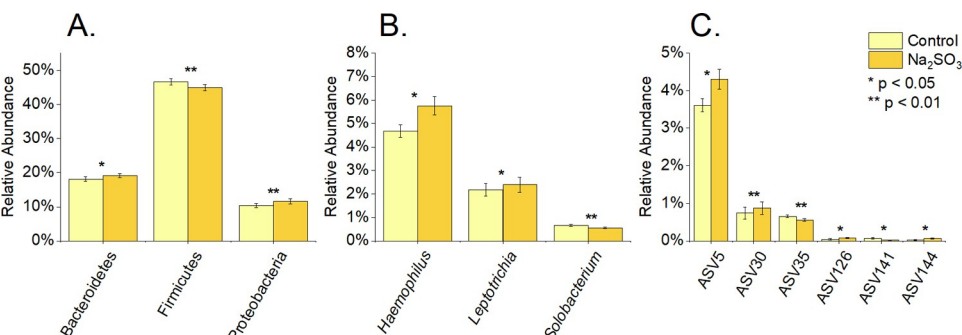

**Fig 6. Differentially abundant bacteria at different taxonomic levels in F2 sample, after sodium sulfite treatment at time 0.** 16S rRNA sequencing was used to determine the % relative abundance of phyla (A), genus (B), and ASV (C) for control and treated samples with $Na_2SO_3$. P values were calculated using paired t-tests.

lysozyme activity (Fig 3). If the population could be measured accurately to show the number of gram-negative vs. gram-positive bacteria present, a relationship might be found due to the greater impact of lysozyme on gram-positive cells. Significant differences in lysozyme activity were observed *between* individuals (Fig 2). Previous studies have corroborated our results of *inter* versus *intra*-individual lysozyme variability [43] and others have shown that lysozyme varies between individuals with respect to circadian rhythm [44].

Preliminary studies on the effects of $Na_2SO_3$ on lysozyme activity in saliva (S-lys) and on human recombinant lysozyme (HR-Lys) indicated a decreased initial rate profile of HR-Lys and to a lesser degree S-lys with sodium sulfite incubation. We hypothesize that this difference may be attributed to the rich mixture of biological components that compose salivary fluid,

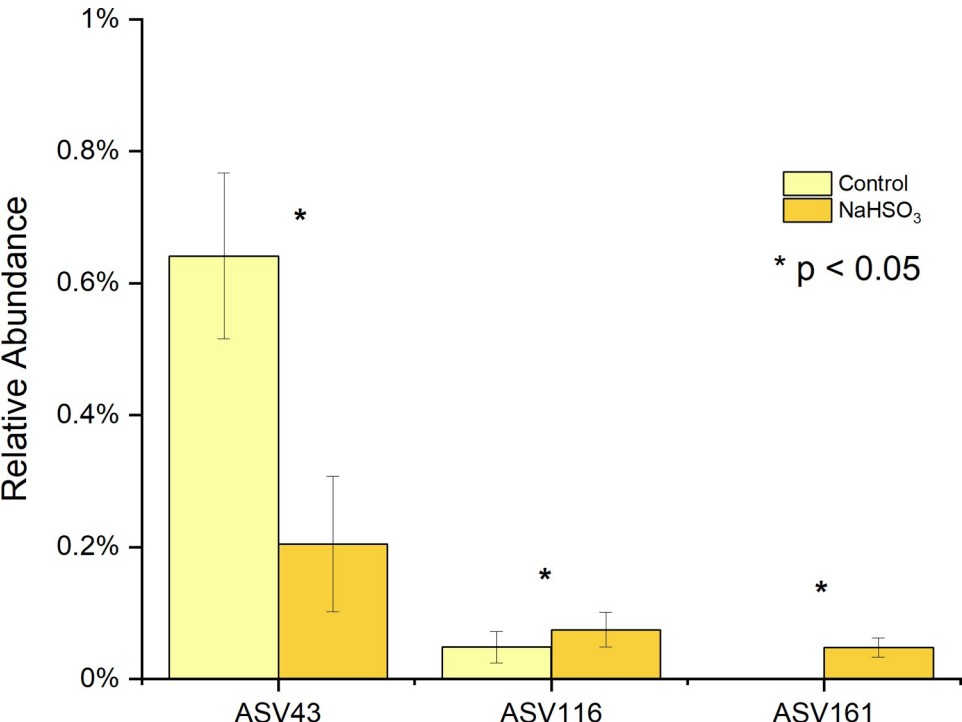

**Fig 7. Differentially abundant ASVs in F2 sample after $NaHSO_3$ treatment at time 0.** ASV43 shows a significant decrease while ASV116 and ASV161 increased significantly after the treatment.

**Table 2. A: Bacteria species/ASVs exhibiting a significant change in relative abundance after exposure to sodium sulfite.** B: Bacteria species/ASVs exhibiting a significant change in relative abundance after exposure to sodium bisulfite.

| Species/ASV# | *Haemophilus parainfluenzae ASV5* | *Solobacterium moorei ASV35* | *Alloprevotella rava ASV126* | *Actinomyces naeslundii ASV141 ASV144* | *Leptotrichia wadei ASV30* |
|---|---|---|---|---|---|
| **Gram Stain** | Negative | Positive | Negative | Positive | Negative |
| **Cellular Respiration** | Facultatively Anaerobic | Obligate Anaerobe | Obligate Anaerobic | Anaerobic or Microaerophilic | Anaerobe |
| **Descriptive** | This type of bacteria is known for its infectious abilities. [32] | Association with Halitosis [33] | Found in mouth in plaque | A is a common bacteria found in plaque. [1] | Normal oral flora |
| **Relative Abundance** | Increased | Decreased | Increased | ASV144 Increased ASV141 Decreased | Increased |
| **Sulfate reducing Bacteria** | No but possess pathways to protect from oxidative stress [32] | No [33] | No [34] | No [1] | Yes [35] |

| Species/ASV# | *Veillonella parvula-tobetsuensis ASV43* | *Prevotella salivae ASV116* | *Actinomyces naeslundii ASV161* |
|---|---|---|---|
| **Gram Stain** | Negative | Negative | Positive |
| **Cellular Respiration** | Obligate Anaerobes | Obligate Anaerobes | Anaerobic or Microaerophilic |
| **Descriptive** | This type of bacteria belongs to the phylum Firmicutes which is a common group that is found in the oral cavity. [36] | This bacteria genus belongs in the phylum Bacteroidetes, a common phylum found in a human GI tract and in periodontal abscesses [5] | This bacterium is a member of the phylum Actinobacteria and is a common bacteria found in plaque. [7] |
| **Relative Abundance** | Increased | Decreased | Decreased |
| **Sulfate Reducing Bacteria** | Yes [36] | Not known | No [7] |

where sulfite may act on bacteria as well as nucleophiles and nonspecific inhibitors [45]. These early studies suggest that sodium sulfite would affect lysozyme function in human saliva and subsequently alter the mouth microbiome. However, we have yet to determine conclusively that sulfite inhibits salivary lysozyme activity and further studies are in progress. More studies are needed to look at the relationship between lysozyme activity, sulfites, and the makeup of the mouth microbiome.

## 16s rRNA sequencing

This study identified five major bacterial phyla (Fig 4) as the most abundant in all control and treated samples. These five phyla, *Actinobacteria*, *Bacteroidetes*, *Firmicutes*, *Fusobacteria*, and *Proteobacteria*, are considered the "core microbiome" in the human mouth since they have been consistently reported as the main bacteria found in both healthy and diseased conditions [1,3,16,46]. However, significant changes in the saliva microbiomes were observed, dependent on the type of sulfite exposure. Alterations of the *Firmicutes/Bacteroidetes* ratio, which is widely accepted to have an important influence in maintaining normal intestinal and mouth homeostasis [1,3,16], was observed with $Na_2SO_3$ but not with $NaHSO_3$ (Fig 5). A more shifted microbial community structure (relative abundance), but only a slight increase in the species richness ($p < 0.10$) was observed in the $Na_2SO_3$ treated samples. However, $NaHSO_3$ samples showed a much higher alpha-diversity ($p < 0.05$) despite showing far fewer changes in relative abundance (Figs 6–8). These results contrast with the similar results found with both sulfites in the combined ATP assays from all individuals (Fig 1). As for beta diversity, the PCA plot (Fig 9) shows consistent changes in the oral microbial communities when comparing the control samples to the $Na_2SO_3$ treated samples, while no similar pattern was found in the $NaHSO_3$ treated samples. These findings, along with our previous studies [24] seem to indicate a difference in the selective potency of these two types of sulfites. One possible explanation for the increase in alpha diversity with both sulfites might be due to the toxic effect on certain

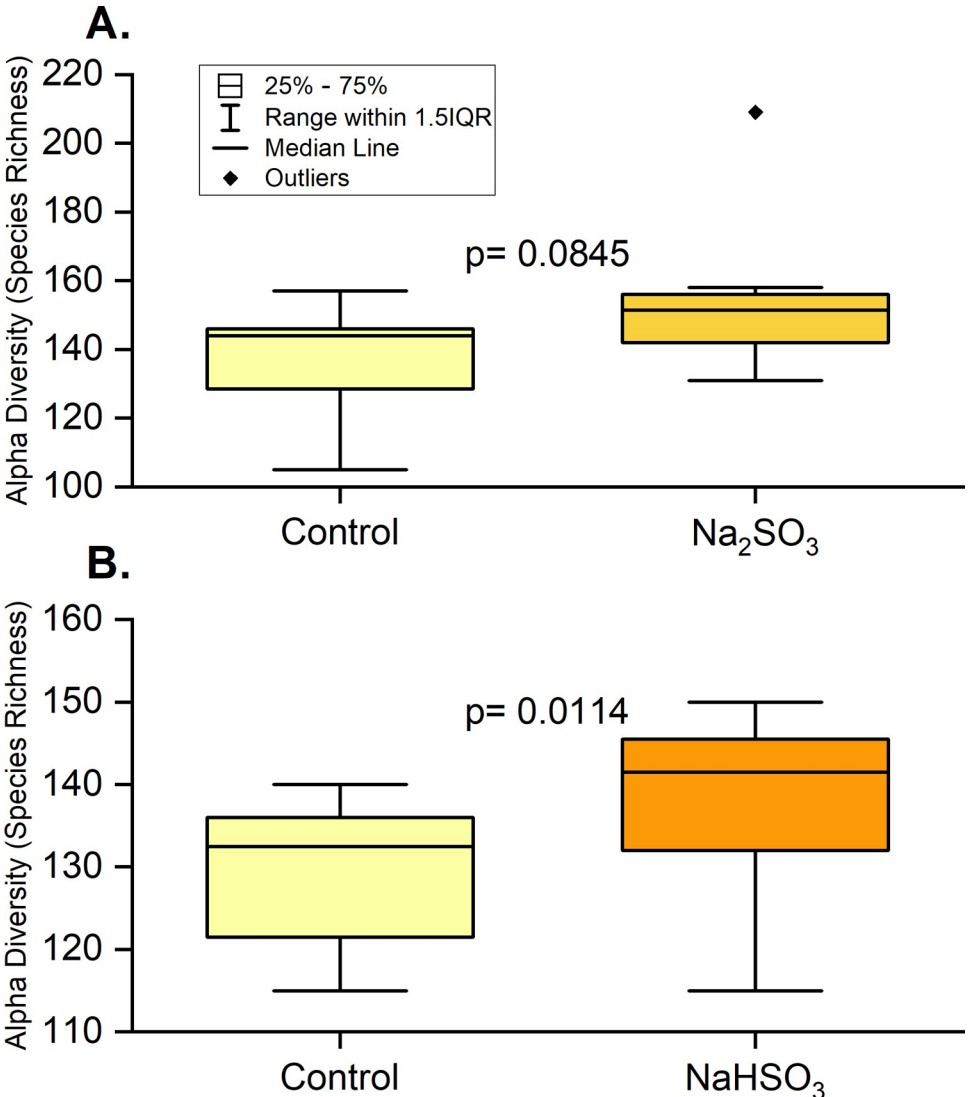

**Fig 8. Alpha diversity in F2 samples with sulfite treatment at time 0.** 16S rRNA sequencing was used to determine species richness (based on relative abundance) for control and treated samples with $Na_2SO_3$ (A) and $NaHSO_3$ (B). A paired t-test showed that species richness increased significantly after treatment with $NaHSO_3$ and was marginally significant after treatment with $Na_2SO_3$.

predominant bacterial species (thereby lowering their relative abundance) that may have then revealed the presence of other species normally present in populations too low to detect [47].

A comparison of the sequencing data to the ATP assays (Fig 4A and 4B) from individual "F2" samples, indicates a significant decrease in cell numbers with both sulfite treatments. However, a greater change was observed with $Na_2SO_3$ vs. $NaHSO_3$. It is interesting to note that a decrease in [DNA] of treated samples from individual "F2" was observed, with an average decrease of 18% (p = 0.34) in sodium bisulfite and 56% (p = 0.05) in the sodium sulfite treated samples (S3 Appendix). While there are many factors that may lead to an observed difference in [DNA], this data supports our observation of a 30% and 40% average decrease in viable cells in respective samples.

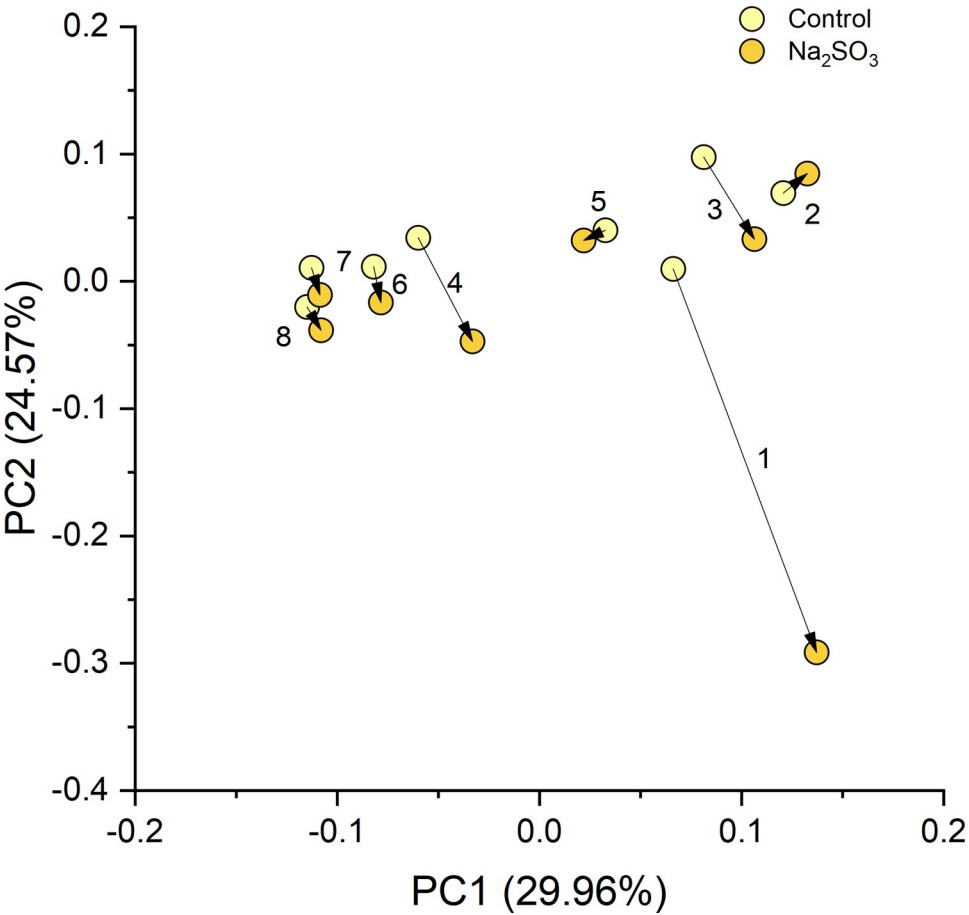

**Fig 9. Beta diversity of F2 Na₂SO₃ treated samples.** The PCA of all the ASVs from the control and Na₂SO₃ treated F2 samples, show consistent directions of changes in the microbial communities under the treatment of Na₂SO₃.

It is well established that DNA from both, recently lysed, as well as living cells may be counted in 16S rRNA gene targeted sequencing [2]. The difference in observed changes (dependent on sulfite type), between our observed sequencing and ATP activity may indicate the quantitative change threshold of viable cells needed in order to detect significance, in this method of sequencing, effectively [47].

## Conclusions

Our results show that sulfites have a clear and significant impact on some bacterium types found in the mouth. We hypothesize that sulfite susceptibility/metabolism differences allow some bacteria found in saliva to increase in numbers while others decrease. Bacteria that produce sulfur nucleotide reductase enzymes (SRB) or have other mechanisms to avoid oxidative stress, are better able to survive, while those that don't will be more susceptible to sulfites toxic effects. The connections of $H_2S$ production from some SRB and its impact on both oral and intestinal inflammation and disease has been previously reported [39,40,42]. Susceptibility of different types of bacteria to lysozyme and the effects of sulfites on lysozyme activity may also be a factor.

To summarize, sulfite food preservatives appear to be affecting the makeup of the mouth microbiome by more than one mechanism including death by oxidative stress to non-SRB bacterial types and an energy source for SRB bacterial types.

Studies on the microbiomes found in or on the human body are complex and in constant flux due to the environment. The use of more than one cell detection method to measure changes in mixed populations of bacteria existing in the mouth is recommended for more accurate and sensitive assessments. ATP tests used to detect changes in the number of viable cells in saliva samples after exposure to either sulfite revealed a significant decrease in cells in all samples, whereas changes identified through 16s rRNA sequencing were less consistent between the samples depending on the type of sulfite tested. Future studies are indicated to include more individual's samples for 16s rRNA sequencing to compare to ATP assays, to examine connections between diseases of the digestive process and the intake of sulfite preservatives.

## Supporting information

**S1 Fig. Baseline lysozyme vs.** ATP activity in F2 control samples.
(TIF)

**S1 File. Supplementary information on methods of treatment.**
(DOCX)

**S1 Appendix. Data corresponding to Figs 1–8.**
(XLSX)

**S2 Appendix. Raw data for statistical applications supporting Fig 1 and Table 1.**
(XLSX)

**S3 Appendix. [DNA] data from 16s sequencing.**
(XLSX)

## Acknowledgments

We would like to extend our appreciation for the helpful comments and review of the manuscript by our colleagues at UH Maui College STEM department including Tom Blamey who helped extensively with our statistical and graphing computations and retired professor Dr. Richard Allen who served as a mentor for students work and a consultant on the protein biochemistry work. We would also like to give our deep appreciation of his time for critical editing and data analysis consultations by Dr. Bret Bessac, Texas Tech University Health Sciences Center.

## Author Contributions

**Conceptualization:** Sally V. Irwin, Peter Fisher, Rachael S. Kent.

**Data curation:** Sally V. Irwin, Luz Maria Deardorff, Youping Deng, Peter Fisher, Michelle Gould, Junnie June, Rachael S. Kent, Fracesca Yadao.

**Formal analysis:** Sally V. Irwin, Luz Maria Deardorff, Youping Deng, Peter Fisher, Michelle Gould, Junnie June, Rachael S. Kent, Yujia Qin, Fracesca Yadao.

**Funding acquisition:** Sally V. Irwin.

**Investigation:** Sally V. Irwin, Luz Maria Deardorff, Peter Fisher, Michelle Gould, Rachael S. Kent, Fracesca Yadao.

**Methodology:** Sally V. Irwin, Peter Fisher.

**Project administration:** Sally V. Irwin, Peter Fisher.

**Resources:** Sally V. Irwin.

**Supervision:** Sally V. Irwin, Peter Fisher, Michelle Gould, Junnie June.

**Validation:** Sally V. Irwin, Youping Deng, Peter Fisher, Junnie June, Rachael S. Kent, Yujia Qin.

**Visualization:** Sally V. Irwin.

**Writing – original draft:** Sally V. Irwin, Michelle Gould, Rachael S. Kent, Fracesca Yadao.

**Writing – review & editing:** Sally V. Irwin, Luz Maria Deardorff, Youping Deng, Peter Fisher, Michelle Gould, Junnie June, Rachael S. Kent, Yujia Qin, Fracesca Yadao.

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
