## [Decision Letter · Decision Letter 0]

9 Dec 2021

PONE-D-21-31327Sulfite preservatives effects on the mouth microbiome: changes in viability, diversity and composition of microbiota.PLOS ONE

Dear Dr. Irwin,

Thank you for submitting your manuscript to PLOS ONE. After careful consideration, we feel that it has merit but does not fully meet PLOS ONE’s publication criteria as it currently stands. Therefore, we invite you to submit a revised version of the manuscript that addresses the points raised during the review process.

Please address all concerns below raised by the reviewers, with particular importance to viability measurements.

We look forward to receiving your revised manuscript.

Kind regards,

Peter Gyarmati

Academic Editor

PLOS ONE

Journal Requirements:

Reviewers' comments:

Reviewer's Responses to Questions

**Comments to the Author**

1. Is the manuscript technically sound, and do the data support the conclusions?

Reviewer #1: Partly

Reviewer #2: Partly

2. Has the statistical analysis been performed appropriately and rigorously? 

Reviewer #1: Yes

Reviewer #2: Yes

3. Have the authors made all data underlying the findings in their manuscript fully available?

Reviewer #1: Yes

Reviewer #2: Yes

4. Is the manuscript presented in an intelligible fashion and written in standard English?

Reviewer #1: Yes

Reviewer #2: Yes

5. Review Comments to the Author

Reviewer #1: This study examines the effects of sulfite preservatives on saliva samples obtained from a limited number of volunteers (10 individuals) using the following endpoints: microbiota diversity and composition, microbiota viability, and lysosyme activity. The results are original and interesting. Several points are unclear and/or not sufficiently discussed.

1. Although the limitation in using the "ATP activity" test is discussed, I still do not really understand this term and what it exactly measured. If I understand correctly, this assay measure ATP in the samples, and if ATP is decreased, it can be presumed that bacterial cells have impaired metabolic activity and/or are dead cells. If this is correct, then it is not an "ATP activity"test, but a measure of ATP in the samples, thus considered as a net balance between ATP synthesis and ATP utilization by bacterial cells. This need to be better explained in terms of what is measured, and what it means in terms of bacterial metabolic activity and viability to make the most correct interpretation.

2. What could be the consequences of the effects of sulfite preservatives on microbiota in terms of oral diseases/dysfunctions. Please discuss.

3. Sulfite preservative are useful preservatives. So please indicate the possible beneficial over deleterious effects ratio of the use of sulfite preservative. What could replace them for preservation of food if needed?

4. Are there any available data indicating conversion of sulfite preservatives into hydrogen sulfide by the oral microbiota?

Reviewer #2: Major concerns:

1. At no point is it described why two different sulfite types were used / unclear what the significance of that was to this particular study.

2. Line 496 – I do not see where this conclusion (an energy source for SRB bacterial types) is directly supported by any data from this paper. In this reviewers opinion this is an appropriate hypothesis but remains speculative and not conclusive.

3. As written this reviewer must assume that all saliva samples were first diluted and then frozen (line 159). Then all downstream experiments are performed on samples that were presumably thawed prior to sulfite exposure. This means that all downstream experiments looking at cell viability and lysozyme activity were performed on samples that underwent at least one freeze-thaw cycle. This reviewer is concerned about how much a prior freeze-thaw cycle would impact the overall viability of all cells in the initial population as well as lysozyme activity pre/post freezing. Did the authors measure viability of fresh samples prior to then after freezing? The concern is that a large decrease in viability may have happened to all samples up front and this study is only assaying the smaller population of survivors which makes one wonder as to the ultimate relevance of the data.

4. How do the sulfite concentrations used in the assays here compare to the presumed sulfite concentrations in the mouth after consuming sulfite-containing food products? Even knowing if they were in the same order of magnitude would be sufficient but this reviewer would like to know how the test conditions compare to the presumed exposure during consumption.

Minor concerns:

1. Line 36 – “American diet” – potentially “North American diet” ?

2. Abstract seems fairly heavy with specific results and data and less of a summary of the entire body of work.

3. Line 67-8 – mentions colonization of aerobes but lists Veillonella which is an anaerobe.

4. Line 159 – participant age / gender recorded but no ethnicity demographics given. Was this included / considered in the metadata for downstream analysis?

5. Line 170 – ppm used for sulfite. Was this the final concentration in the saliva sample for each? Might be nice to give the Molarity for each in the final saliva sample.

6. Line 173-4 RPM given but g values not, it is unclear if this was sufficient centrifugation to reliably pellet all cells in a saliva sample reproducibly.

7. Line 217 – any rationale for using V3-V4 for oral samples? M. Eren et al PNAS 2014 (Oligotyping analysis of the human oral microbiome) indicates that V1-V3 better discerns oral species than V3-V5.

8. Line 430-1 – This might be somewhat addressed by looking at the relative abundance of each Gram positive phyla vs lysozyme status compared to Gram negative phyla vs lysozyme amounts.

9. Line 468-470 – this could also be sequencing of reagent / aqueous contaminants that show up in amplification of low-template abundance samples. Do these samples look more similar to the reagent control samples sequenced?

6. PLOS authors have the option to publish the peer review history of their article (what does this mean?). If published, this will include your full peer review and any attached files.

Reviewer #1: No

Reviewer #2: No

---

## [Author Response · Author response to Decision Letter 0]

22 Jan 2022

Reviewer #1: This study examines the effects of sulfite preservatives on saliva samples obtained from a limited number of volunteers (10 individuals) using the following endpoints: microbiota diversity and composition, microbiota viability, and lysosyme activity. The results are original and interesting. Several points are unclear and/or not sufficiently discussed.

1. Although the limitation in using the "ATP activity" test is discussed, I still do not really understand this term and what it exactly measured. If I understand correctly, this assay measure ATP in the samples, and if ATP is decreased, it can be presumed that bacterial cells have impaired metabolic activity and/or are dead cells. If this is correct, then it is not an "ATP activity” test, but a measure of ATP in the samples, thus considered as a net balance between ATP synthesis and ATP utilization by bacterial cells. This need to be better explained in terms of what is measured, and what it means in terms of bacterial metabolic activity and viability to make the most correct interpretation.

The amount of ATP in each saliva sample is measured based on the luminescence produced due to the chemical reaction of the luciferase enzyme and luciferin present in the BacTiter-Glo reagent. We refer to this as “ATP activity” in that it is the direct measurement of ATP driving the luciferase reaction, however it is also considered a measure of the metabolically active (viable cells) in the samples that provide the ATP (when lysed) for the reaction, and therefore the amount of ATP in the samples. Revisions for clarity can be found in line # 130-136 in the revised manuscript. The concentration of adenosine triphosphate (ATP) present in living cells can be quantified using a proportional luminescent signal to observe immediate changes in the numbers of viable bacterial cells present in a sample. Sensitivity as low as 0.0001nM ATP, reflecting the test population, is high due to the rapid loss of ATP in non-living cells [25]. This method implements a recombinant luciferase enzyme to oxidize luciferin in the presence of ATP and oxygen to produce oxyluciferin and light.

2. What could be the consequences of the effects of sulfite preservatives on microbiota in terms of oral diseases/dysfunctions. Please discuss.

Changes in the mouth microbiome and diseases related to these changes are still in the early stages of discovery. This is the first study (that we are aware of) that looks at the effects of a food preservative on the mouth microbiome. However, there have been several recent studies showing changes in the mouth microbiome associated with and contributing to diseases found in the oral cavity, gastrointestinal system, liver cirrhosis, certain cancers, endocrine system diseases like diabetes and obesity, some diseases of the nervous system, and cardiovascular disease. These studies were reviewed in a 2018 paper we cited [1] on line #’s 80-84 in the revised manuscript and similar studies which were reviewed in a 2017 paper [10] that also spoke to the possible diagnostic potential of the state of the mouth microbiota and certain diseases.

3. Sulfite preservative are useful preservatives. So please indicate the possible beneficial over deleterious effects ratio of the use of sulfite preservative. What could replace them for preservation of food if needed?

Sulfites have been utilized as a food preservative going back to the Roman days. They have been found to be very effective in their role to limit bacterial growth in many different food types. However, the effects of sulfites on human health has been noted many times over the past 60 years and have led to numerous changes in the regulations of sulfites in food. This was discussed in more detail in our first paper on sulfites effects on 4 probiotic bacteria [24] . Some studies have shown sulfites to be dangerous to humans when ingested at levels as low as 1 ppm [1,3–7]. Due to insufficient statistical data regarding individual sensitivities and consumer intake levels [8,9], it has been difficult to identify the exact level at which these preservatives become harmful. Reactions can occur between these additives and primary constituents naturally present in food, as well as during preparation and digestion, contributing to this conundrum.

Our studies did not look at alternatives to sulfites for food preservation however there are a number of preservatives that we have performed preliminary testing on (but not published) that do not show the toxicity that sulfites have on the mouth bacteria (based on ATP studies). Some herbs, hypertonic environments with high salt or sugars and lowered pH using citric acid or vinegar along with fermentation are all methods of food preservation that are less likely to disrupt the microbiome of the mouth or gut. 

4. Are there any available data indicating conversion of sulfite preservatives into hydrogen sulfide by the oral microbiota?

The following two references from our manuscript describe the common presence of sulfate reducing bacteria in the mouth and the conversion of sulfates to hydrogen sulfide [39] and the reduction of sulfites to sulfates (in the gut) [40]. The presence of sulfite reducing bacteria in the mouth has not been confirmed to our knowledge. 

From reference #39

“In this context, an increased incidence of sulfate-reducing bacteria (SRB) in the oral cavity has been found, which are a part of the common microbiome of the mouth.”

“The amount of SRB in the oral cavity is limited by the amount of sulfate available. Potential sources of sulfate in the subgingival region include free sulfate in the pocket fluid and glycosaminoglycans and sulfur-containing amino acids (cysteine and methionine) from periodontal tissues. SRB then metabolize the sulfate to H2S.” 

From reference #40

“Another essential reaction of the process is sulfite reduction, resulting in the product of APS reduction [4,44]. Sulfite reduction is catalyzed by dissimilatory sulfite reductase (EC 1.8.99.1). This enzyme reduces sulfite to sulfate [44]. Sulfide reductase also plays an important role in the process of assimilatory sulfate reduction due to sulfide ion production. These sulfides are part of amino acids containing sulfur, such as methionine and cysteine. SRB may have several types of sulfide reductases that can be used for identification. These reductases are desulfoviridin, desulforubin, desulfofuscidine, and protein P582.”

“Sulfite reduction is the last reaction in the process of DSR. Reactive sulfite is converted into toxic sulfide, and then it is released out of the bacterial cell. This reaction is catalyzed by the enzyme sulfite reductase.” 

Eukaryotes, including mammals, have sulfite oxidase in the mitochondria of all their cells. This enzyme oxidizes sulfites to sulfates [Andrei V. Astashkin, in Methods in Enzymology, 2015]. It has been established that there are individuals with deficiencies in this enzyme [Mellis, January 2021] that can lead to severe reactions to sulfites and even with normal levels of the enzyme, the ability of the enzyme to keep up with high levels of sulfites appears to be challenged. There have been studies that have observed variable levels of sulfite oxidase gene expression dependent on the specific tissues in humans and rats [Woo et al., 2003], however we have not been able to find a study that specifically looks at levels produced in the mouth. 

Reviewer #2: Major concerns:

1. At no point is it described why two different sulfite types were used / unclear what the significance of that was to this particular study.

The reason we looked at the effects of both sodium sulfite and sodium bisulfite at the concentrations indicated was based on the results from our previous paper that tested the effects of these two types of preservatives on four different beneficial bacteria species. In this study [24] our results are summarized here: 

“All three Lactobacillus species stopped increasing in number within 2 hrs of exposure at concentrations between 250–750 ppm NaSO3, (Fig 2), and were found to be non-viable within 4 hours of exposure, at concentrations ranging between 1000 -3780ppm, dependent on species tested (Table 3). Streptococcus thermophilus also stopped increasing in number within two hours of exposure at concentrations between 250–500ppm; however, the bacteria were still viable in all concentrations of sodium sulfite tested up to 6 hours exposure time. These results indicate a bacteriostatic effect from sodium sulfite on all bacteria within two hours of exposure and a bactericidal effect on all the Lactobacillus species by 4 hours of exposure

Sodiium bisulfite: All bacteria stopped increasing in cell number within two hours of exposure at sodium bisulfite concentrations between 250–500 ppm NaHSO3, (Fig 3). Sodium bisulfite was observed to be bactericidal at 2 hours exposure to L. casei and L. rhamnosus at 1000ppm. Lactobacillus plantarum was found to be non-viable by 4 hours exposure at ≥ 1000ppm. Sodium bisulfite was bactericidal to S. thermophilus at 6 hours exposure and ≥ 1000 ppm.” 

This previous study showed us that the effects of sulfites varied depending on the bacteria genus and species and the type and concentration of the sulfite tested. This was discussed briefly in the introduction (line # 113-116 revised manuscript) and in the discussion (line # 401-406 revised manuscript). Our current study was done to observe the effects of sulfites on communities of bacteria found in saliva, to assess an environment closer to one that is “in-vivo”. We have added some additional information to those sections to make this information clearer to the reader. 

2. Line 496 – I do not see where this conclusion (an energy source for SRB bacterial types) is directly supported by any data from this paper. In this reviewer's opinion this is an appropriate hypothesis but remains speculative and not conclusive.

We are in agreement that the idea that sulfites may be serving as an energy source for some SRB bacteria should still be considered a hypothesis rather than a conclusion. Our statement was based on work from others (citation #39 and see response to reviewer 1’s question #4) and our observations of changes in relative abundance of 9 ASV’s after treatment (line # 344-350 revised manuscript and Table’s 2A and 2B).

Nine different anaerobic ASV’s exhibited significant changes in relative abundance with sulfite treatments. Four out of the five that showed an increase in abundance were Gram negative and three out of those five were sulfite reducing bacteria (SRB) or have the ability to avoid oxidative stress. Four ASV’s decreased in relative abundance with sulfite treatments. Three out of the four were Gram positive and one out of the 4 was a SRB.

The common presence of SRB bacteria in the mouth has been established by others and our studies indicated a trend towards increasing abundance in SRB’s and a decrease in abundance of non-SRB’s. Line # 486-500 in the revised manuscript has been updated to make it more clear that our statement is a hypothesis rather than a conclusion.

3. As written this reviewer must assume that all saliva samples were first diluted and then frozen (line 159). Then all downstream experiments are performed on samples that were presumably thawed prior to sulfite exposure. This means that all downstream experiments looking at cell viability and lysozyme activity were performed on samples that underwent at least one freeze-thaw cycle. This reviewer is concerned about how much a prior freeze-thaw cycle would impact the overall viability of all cells in the initial population as well as lysozyme activity pre/post freezing. Did the authors measure viability of fresh samples prior to then after freezing? The concern is that a large decrease in viability may have happened to all samples up front and this study is only assaying the smaller population of survivors which makes one wonder as to the ultimate relevance of the data.

Your assumption is correct in that the samples were frozen and thawed prior to sulfite exposure. The details of this process and the possible inherent errors due to this protocol is discussed in the “Supporting information”. In preliminary studies, we tested fresh vs. frozen and thawed samples of saliva as well as pelleted and resuspended in PBS and those that were only treated and then ATP tested. The results trended towards a slight reduction in the number of cells which we attributed primarily to pipetting errors. However, we hypothesized that the loss of cells to pipetting errors or the freeze thaw cycle would be similar between controls and treated samples. This is where relatively large numbers of samples were important to statistically verify the trends we observed as a decrease in cell viability with sulfite treatments.

In addition to the study presented here, we initially assayed 5 individuals over a 3 month period (12 samples/person). In this set of experiments, some samples from the same individual were treated on different days, and ATP tested (sometimes) in different assays. Despite these variables, the results from these experiments showed almost identical results in the decrease of viable cells with treatment compared to our current study, which treated all of the samples from any one individual on the same day and included all ATP testing in the same assay. In addition to the ATP data in the preliminary study, samples were tested for growth by observing OD600 readings over a 6.5 hour period. We observed an initial decrease in cell numbers in the treated which remained lower than controls throughout study, and an increase in cell numbers in the controls. This preliminary data was discussed briefly in the discussion section in line #’s 419-427. 

Lastly the [DNA] of the sequenced samples supported our ATP findings in showing a decrease in [DNA] in the treated vs. control samples (line #’s 475-478 and appendix S3 in revised manuscript).

There are also several other studies that have utilized -80℃ or - 20℃ freezing of saliva samples with later thaws and testing [5, 6, 11, 12, 43], some with more than one freeze thaw cycle. It is not ideal but we feel that it does still allow for the comparison between treated and untreated samples even with the inherent variabilities from this type of protocol.

4. How do the sulfite concentrations used in the assays here compare to the presumed sulfite concentrations in the mouth after consuming sulfite-containing food products? Even knowing if they were in the same order of magnitude would be sufficient but this reviewer would like to know how the test conditions compare to the presumed exposure during consumption.

This is the million dollar question. As a first step we can test levels that are allowable in food (which we did) and see if there are any significant effects observed. In our previous study (see answer to question 1, reviewer 2), we observed bacteriostatic and bactericidal effects at much lower concentrations of sulfites then tested here but with longer exposure times. We tested 6 other common food preservatives in preliminary experiments at GRAS levels and have not observed any to have as significant of an effect on the microbiota found in saliva. In fact, many have had no effect (i.e. Nitrites) and some have actually increased the number of cells in saliva based on ATP (i.e. Methyl Parabans). 

Other studies have looked at the likelihood of people ingesting more than the allowable levels on a regular basis and have found that it is a common occurrence. In one study by Leclercq et al. “It was shown that the diets obtained from these foods would lead to an intake of 23mg/day in children and 50mg/day in adults (both slightly above the ADI for respectively a 30kg child and a 60kg adult). Among all sulphite-containing foods, the highest contributors to the intake were dried fruit and wine, both ingested without further treatment. The analysis of specific consumption data confirmed the existence of a risk of exceeding the ADI related to sulphite residue levels in wine.”

The problems faced when trying to determine the “safe” amount of a food preservative for a population is summarized by Fazio and Warner here: 

“The fate of added sulfites is highly dependent on the chemical nature of the food, the type and extent of storage conditions, the permeability of the package and the level of addition. The combination with organic constituents, the equilibrium between the various inorganic forms, the volatilization of sulfur dioxide and the oxidation to sulfates are all important reactions, and their relative importance will depend mostly on the food involved.” (Fazio & Warner,1990)

Minor concerns:

1. Line 36 – “American diet” – potentially “North American diet” ? 

Changed in revised manuscript.

2. Abstract seems fairly heavy with specific results and data and less of a summary of the entire body of work. 

Agreed, however with the 300 word limit it was difficult to present the experiment and results in a manor generally required within this limited word count. 

3. Line 67-8 – mentions colonization of aerobes but lists Veillonella which is an anaerobe. 

This was not intended, it has been corrected in revised manuscript (line #68).

4. Line 159 – participant age / gender recorded but no ethnicity demographics given. Was this included / considered in the metadata for downstream analysis? 

It was not included or considered for this data set. 

5. Line 170 – ppm used for sulfite. Was this the final concentration in the saliva sample for each? Might be nice to give the Molarity for each in the final saliva sample. 

Added in lines 169-171 in revised manuscript.

6. Line 173-4 RPM given but g values not, it is unclear if this was sufficient centrifugation to reliably pellet all cells in a saliva sample reproducibly. 

This was a typo that should have said 4600 RCF’s rather than RPM’s. We corrected this in the revised manuscript and added the equivalent in RPMs which is 7000 RPMs (line # 173-174 in revised manuscript). 

7. Line 217 – any rationale for using V3-V4 for oral samples? M. Eren et al PNAS 2014 (Oligotyping analysis of the human oral microbiome) indicates that V1-V3 better discerns oral species than V3-V5. 

We were unaware of the research showing a higher degree of specificity using V1-V3 for the oral microbiome. We went with the V3-V4 region based on other recent papers on the microbiome (2,5) and from the advice of Zymo, the company that performed the sequencing of our samples. However, we are encouraged by the similarities in the results of both sequencing and ATP assays for individual F2. 

8. Line 430-1 – This might be somewhat addressed by looking at the relative abundance of each Gram positive phyla vs lysozyme status compared to Gram negative phyla vs lysozyme amounts. 

This would be interesting to look at with more sequencing data. In our sequencing data which is from one individual (32 samples) we see no change in the relative abundance of gram positive or gram negative bacterial types with sodium bisulfite treatment, however we do see a 6% decrease in gram positive and a 6% increase in gram negative bacterial types with sodium sulfite treatment (based on phylum data). 

9. Line 468-470 – this could also be sequencing of reagent / aqueous contaminants that show up in amplification of low-template abundance samples. Do these samples look more similar to the reagent control samples sequenced?

We are not completely sure what is being asked/commented on here. From the manuscript “One possible explanation for the increase in alpha diversity with both sulfites might be due to the toxic effect on certain predominant bacterial species (thereby lowering their relative abundance) that may have then revealed the presence of other species normally present in populations too low to detect [47].”

Our samples did not look like the controls used by Zymo and the negative controls were very clean. In the bisulfite treated and control samples the majority (94.24%) of the sequences were classified into species level and for the sodium sulfite treated and control samples 93.84% of the sequences were classified into the species level. 

From Zymo:

For projects that included DNA purification from raw samples, the ZymoBIOMICS® Microbial Community Standard was used as positive control; a blank extraction sample was used as a negative control.

---

## [Decision Letter · Decision Letter 1]

28 Feb 2022

Sulfite preservatives effects on the mouth microbiome: changes in viability, diversity and composition of microbiota.

PONE-D-21-31327R1

Dear Dr. Irwin,

We’re pleased to inform you that your manuscript has been judged scientifically suitable for publication and will be formally accepted for publication once it meets all outstanding technical requirements.

Kind regards,

Peter Gyarmati

Academic Editor

PLOS ONE

Additional Editor Comments (optional):

Reviewers' comments:

Reviewer's Responses to Questions

**Comments to the Author**

1. If the authors have adequately addressed your comments raised in a previous round of review and you feel that this manuscript is now acceptable for publication, you may indicate that here to bypass the “Comments to the Author” section, enter your conflict of interest statement in the “Confidential to Editor” section, and submit your "Accept" recommendation.

Reviewer #1: All comments have been addressed

2. Is the manuscript technically sound, and do the data support the conclusions?

Reviewer #1: Yes

3. Has the statistical analysis been performed appropriately and rigorously? 

Reviewer #1: Yes

4. Have the authors made all data underlying the findings in their manuscript fully available?

Reviewer #1: Yes

5. Is the manuscript presented in an intelligible fashion and written in standard English?

Reviewer #1: Yes

6. Review Comments to the Author

Reviewer #1: All my comments have been taken into consideration, and for some of them used in order to revise the manuscript. The responses to the questions are adequate.

7. PLOS authors have the option to publish the peer review history of their article (what does this mean?). If published, this will include your full peer review and any attached files.

Reviewer #1: No

---

## [Editor Report · Acceptance letter]

7 Mar 2022

PONE-D-21-31327R1 

Sulfite preservatives effects on the mouth microbiome: changes in viability, diversity and composition of microbiota. 

Dear Dr. Irwin:

I'm pleased to inform you that your manuscript has been deemed suitable for publication in PLOS ONE. Congratulations! Your manuscript is now with our production department. 

Kind regards, 

on behalf of

Dr. Peter Gyarmati 

Academic Editor

PLOS ONE